# “Smart” Triiodide Compounds: Does Halogen Bonding Influence Antimicrobial Activities?

**DOI:** 10.3390/pathogens8040182

**Published:** 2019-10-10

**Authors:** Zehra Edis, Samir Haj Bloukh, Hamed Abu Sara, Hanusha Bhakhoa, Lydia Rhyman, Ponnadurai Ramasami

**Affiliations:** 1College of Pharmacy and Health Science, Ajman University, Ajman P.O. Box 346, UAE; s.bloukh@ajman.ac.ae (S.H.B.); h.abusara@ajman.ac.ae (H.A.S.); 2Computational Chemistry Group, Department of Chemistry, Faculty of Science, University of Mauritius, Réduit 80837, Mauritius; hbhakhoa@gmail.com (H.B.); lyd.rhyman@gmail.com (L.R.); 3Department of Chemical Sciences, University of Johannesburg, Doornfontein, Johannesburg 2028, South Africa

**Keywords:** halogen bonding, triiodide, antimicrobial activity, DFT, lipophilicity, SEM, EDS

## Abstract

Antimicrobial agents containing symmetrical triiodides complexes with halogen bonding may release free iodine molecules in a controlled manner. This happens due to interactions with the plasma membrane of microorganisms which lead to changes in the structure of the triiodide anion. To verify this hypothesis, the triiodide complex [Na(12-crown-4)_2_]I_3_ was prepared by an optimized one-pot synthesis and tested against 18 clinical isolates, 10 reference strains of pathogens and five antibiotics. The antimicrobial activities of this symmetrical triiodide complex were determined by zone of inhibition plate studies through disc- and agar-well-diffusion methods. The triiodide complex proved to be a broad spectrum microbicidal agent. The biological activities were related to the calculated partition coefficient (octanol/water). The microstructural analysis of SEM and EDS undermined the purity of the triiodide complex. The anionic structure consists of isolated, symmetrical triiodide anions [I-I-I]^−^ with halogen bonding. Computational methods were used to calculate the energy required to release iodine from [I-I-I]^−^ and [I-I···I]^−^. The halogen bonding in the triiodide ion reduces the antibacterial activities in comparison to the inhibitory actions of pure iodine but increases the long term stability of [Na(12-crown-4)_2_]I_3_.

## 1. Introduction

Iodine and some polyiodides are widely used as microbicides and disinfectants against pathogenic microorganisms [1]. Polyiodide anions can be built through the attachment of iodine molecules to iodide ions and incorporated next to large cations into crystalline solids [2,3,4,5,6]. They contain [I_2k+n_]^n-^ units, which are formed through donor-acceptor interactions of the combined iodine molecules and iodide ions. Raman spectroscopy and electron density calculations are used to identify these interactions which range from covalent bonds within the I**_2_**-units to halogen bonding within three-center systems of [I-I-I]^−^ and [I-I···I]^−^ present in polyiodides [2,5,6]. Within the last few years, new polyiodide structures emerged demonstrating a positive effect of crown ethers and metal cations on their structures and stability [6]. Anion-π interactions in combination with electron-poor, polarized salts result in stable polyiodides in the solid-state [7]. Polyiodides of the type I_3_^–^∙∙∙I_3_^−^can also be stabilized via weak forces of interaction such as hydrogen bonding [8]. The I–I distances in the I_3_^−^anion and the I_3_^–^∙∙∙I_3_^−^interactions in its dimers were calculated theoretically in the gas phase [9]. 

There is a need to find new antimicrobial compounds due to the increasing number of infectious diseases caused by pathogenic microorganisms, the widespread use of antibiotics and the emergence of resistance towards antibiotics [10]. Many new strains of the known bacterial pathogens are highly multidrug-resistant [11]. The increase in bacterial infection-associated morbidity and mortality worldwide through drug and multidrug-resistant bacterial strains is a danger for humankind [12,13]. 

Inorganic antimicrobial agents, especially iodine, are marketed as an ingredient in products due to their biocidal properties [1,6,14,15]. Iodine is also used to control biohazards in stockpiles of biological weapons [16,17]. Iodine-rich, but thermally stable compounds release I_2_ upon deformation to inactivate the microorganisms. As an element, iodine is not suitable as an ingredient as a defeat agent due to its sublimation [16,17]. The same problem reduces its durability, thermal stability and long-term effectiveness when used in other products as an antimicrobial agent. However, the advantages of using iodine are due to it being essential to human health, nontoxic in mammals and not causing substantial environmental pollution [16,17]. 

Metal cations, transition metals [18,19,20,21,22] and polymers have also gained interest in drug development and biocidal applications [23,24,25,26]. Povidone-iodine (polyvinylpyrrolidone-I_2_ or PVP-I_2_) is an effective disinfectant [6,27]. Free molecular iodine is released from the complex PVP-I_2_ and inhibits bacterial growth by known mechanisms [6,27]. The main disadvantage of this compound is its hydrophilic nature, limiting its use and durability as an antimicrobial agent. Gao et al. synthesized PVP-I_2_ nanoparticles with strong antibacterial activities against *E. coli* and *S. aureus* [27]. Hydrophobic antimicrobial compounds with long-term effectiveness, thermal stability and suppressed iodine sublimation will be promising future solutions.

Polymeric antibacterial agents have the ability to increase the long-term effectiveness and the stability of the biocidal agent [26,28]. Iodine in combination with naturally occurring polymeric biocides like chitosan and starch are toxic against microorganisms [2,26,28]. Long-term effectiveness and stability of the biocidal agent are important requirements. These can be achieved by using polyiodide-structures as a complex, molecular backbone, releasing the biocidal agent I**_2_** slowly and effectively over a long period of time [28]. Previous investigations of our group show the stability and structure of such compounds [29,30]. The linear, symmetrical triiodide anions in [Na(12-crown-4)_2_]I_3_ [29] are the perfect example of a three-center-system [I-I-I]^−^ with halogen bonding [5]. The Raman stretching vibration appears at 108 cm**^-1^** due to the electron density redistribution during the formation of halogen bonds [5,6,29]. The iodine atoms are highly involved in the halogen bonding, while the covalent character of I**_2_** units is reduced. Therefore, there is a shift of the usual reference band in iodine crystal from 180 to 108 cm**^−1^**. The latter corresponds to the symmetric vibration of triiodide anions. The strong involvement of the triiodide ion in halogen bonding with the weakened covalent bonding character is expected to reduce the antimicrobial action of the anionic structure due to higher stability. The antimicrobial activity of polyiodides depends on the release of free molecular iodine (I_2_) from the stable polyiodide units [6,27]. I_2_ penetrates the plasma membrane of microorganisms and directly attacks the proteins [6,31,32]. Investigations on generally membrane-impermeable metals (sodium, magnesium, calcium, potassium) showed only a mild effect for sodium and no effects for magnesium, calcium or potassium [24].

This study aimed to optimize the synthesis of [Na(12-crown-4)_2_]I_3_ and understand the relation between the anionic structure and antimicrobial activity. Therefore, we tested the triiodide compound against clinical and reference samples in comparison to I_2_, NaI, 12-crown-4 and commonly used antibiotics. The triiodide complex [Na(12-crown-4)_2_]I_3_ was synthesized by an optimized one-pot synthesis and microstructural analysis was performed to examine its purity and surface morphology. The complex was tested against a total of 18 clinical isolates and reference strains of 10 different pathogens in comparison to five antibiotics. The selected Gram-positive cocci were *Streptococcus pneumoniae***,**
*Staphylococcus aureus,*
*S. aureus ATCC 25923,*
*Streptococcus pyogenes, S. pyogenes ATCC 19615, Enterococcus faecalis*, *E. faecalis ATCC 29212* and the spore-forming bacteria *Bacillus subtilis.* The Gram-negative bacteria were *Proteus mirabilis*, *P. mirabilis ATCC 29906*, *Pseudomonas aeruginosa, P. aeruginosa WDCM 00026, Escherichia coli, E. coli WDCM 00013, Klebsiella pneumoniae* and *K. pneumoniae WDCM 00097*. The fungi were *Candida albicans* and its reference strain *C. albicans WDCM 00054*. The antimicrobial activities determined by zone of inhibition plate studies through disc- and agar-well-diffusion methods. [Na(12-crown-4)_2_]I_3_ acted as a strong antifungal and broad-spectrum bacteriocidal agent. We also used computational methods to investigate the relationship between biological activity and partition coefficient between octanol and water (log P_o/w_) [33] as well as the energy required to release iodine from [I-I-I]^−^ and [I-I···I]^−^. 

## 2. Results

The antimicrobial activities of [Na(12-crown-4)_2_]I_3_, I_2_, 12-crown-4 and NaI depend on their molecular interaction with the cellular structure of the selected microorganisms. Iodine acted as the most potent antimicrobial on all clinical and reference microbial strains while [Na(12-crown-4)_2_]I_3_ showed mixed results due to its strong halogen bonding character impairing the release of free molecular iodine from the triiodide unit. The hydrophobicity of this compound was expected to enhance its antimicrobial activities and possible applications. The linear, symmetrical triiodide ions with strong halogen bonding hinder the release of free molecular iodine unless the anionic structure is deformed due to interactions of the complex with microbial membranes. At the same time, the triiodide complex is hydrophobic and durable in contrast to many antimicrobial agents, which incorporate polyiodides or iodine. Our compound [Na(12-crown-4)_2_]I_3_ remains stable for one year in a tightly closed, brown borosilicate reagent bottle stored at 25 °C in darkness. Halogen bonding seems to be a very important characteristic, which increases the stability of polyiodides by preventing the release of iodine.

### 2.1. Antimicrobial Testing by Agar-Well and Disc-Diffusion Studies

[Na(12-crown-4)_2_]I_3_ was highly effective against the Gram-positive bacteria with zones of inhibition (ZOI) between 20 to 43 mm in diameter. Our compound highly inhibited *S. aureus* (ZOI = 43 mm) followed by *S. pyogenes* (34 mm), *E. faecalis* (39 mm) and *S. pneumoniae* (28 mm) (Table 1, Figure 1a). Only the clinical isolate *B. subtilis* remained with an intermediate result of 15 mm (Figure 1a,b). The Gram-negative bacteria *E. coli* (ZOI = 23 mm), *P. aeruginosa* (20 mm) and *K. pneumoniae* (15 mm) were most sensitive towards [Na(12-crown-4)_2_]I_3_ (Table 1, Figure 1b). The others were inhibited strongest by iodine, then NaI and finally [Na(12-crown-4)_2_]I_3_ in the disc diffusion studies. [Na(12-crown-4)_2_]I_3_ has the strongest inhibitory effect on the clinical sample of *C. albicans* with ZOI = 50 mm at a concentration of 13.3. mg/mL. The reference strain *C. albicans WDCM 00054* was inhibited also strongly (40 mm) at a concentration of 10 mg/mL.

[Na(12-crown-4)_2_]I_3_ exerts strong antifungal activity on clinical isolate and reference strains of *C. albicans* in well and disc diffusion methods with the highest inhibition zones compared to all other bacterial strains in this study (Figure 1b). In comparison, the Gram-negative pathogens showed higher resistance towards [Na(12-crown-4)_2_]I_3_ (Table 1, Figure 1b). 

In disc diffusion studies, all the reference microbial strains are less sensitive to [Na(12-crown-4)_2_]I_3_ than the clinical isolates except for *S. aureus ATCC 25923* and *C. albicans WDCM 00054*. These two reference strains exhibited higher inhibition zones than the clinical isolates (Table 1, Figure 1a,b). The Gram-positive bacteria and *C. albicans* are all resistant against NaI except *B. subtilis* and the reference strain *E. faecalis ATCC 29212* although they are strongly inhibited by [Na(12-crown-4)_2_]I_3_ (Table 1). This proves, that the inhibitory action in [Na(12-crown-4)_2_]I_3_ is not caused by iodide ions, but the release of free molecular iodine leaving the [I-I-I]^−^ system and thus, acting as an antimicrobial agent. The yeast *C. albicans* and some Gram-positive bacteria are clearly resistant to 12-crown-4 in disc diffusion studies, while Gram-negative bacteria exhibited ZOI around 9–14 mm in diameter (Table 1). These results prove, that Gram-positive bacteria and the two strains of *C. albicans* (the clinical isolate and the reference strain) are resistant to 12-crown-4 in well and disc diffusion studies although they are strongly inhibited by [Na(12-crown-4)_2_]I_3_. The Gram-negative pathogens have higher susceptibility towards 12-crown-4 compared to the Gram-positive bacteria, but still, they are more sensitive towards [Na(12-crown-4)_2_]I_3_. As a result, the antimicrobial action of [Na(12-crown-4)_2_]I_3_ is related to the release of free molecular iodine, not due to 12-crown-4. The microorganisms proved to be resistant in higher dilution series to ethanol or ethanol/water (1:9) with the [Na(12-crown-4)_2_]I_3_ concentrations of 0.529 mM (0.4 mg/L) and 0.0529 mM (0.04 mg/L). The pathogens showed resistance in all the negative controls by diffusion method using only ethanol, methanol and ethanol/water. The antibacterial activity of diluted [Na(12-crown-4)_2_]I_3_ against Gram-positive and Gram-negative bacteria in comparison to few selected antibiotics by disc diffusion assay is listed in Table 2 and shown in the Figure 2 and Figure 3.

In the dilution series with 13 mg/mL, selected Gram-positive bacteria are inhibited by [Na(12-crown-4)_2_]I_3_ stronger than by gentamicin and erythromycin (Figure 2a), while most of the selected Gram-negative species were more sensitive against commonly used antibiotics (Figure 2b and Figure 3b). Antibiotics showed higher inhibition zones in all cases compared to [Na(12-crown-4)_2_]I_3_ in dilution series of 10 mg/mL, except in the case of the clinical isolate of *S. pyogenes* and erythromycin (Figure 2). Amikacin (ZOI = 22 mm) and the triiodide compound (ZOI = 23 mm) which exhibit similar inhibition zones against *E. coIi* (Table 2, Figure 2b and Figure 3b).

### 2.2. Microstructural Analysis

The microstructural properties (morphology, size, distribution, surface state and elemental mapping) of [Na(12-crown-4)_2_]I_3_ were analyzed by a scanning electron microscope (Quanta SEM) at 5 and 10 kV with low and high vacuum mode. The results regarding surface morphology, composition details and elemental energy-dispersive X-ray spectroscopic (EDS) information are in agreement with previous findings obtained through different analytical methods [29]. Figure 4 and Figure 5 illustrate the micrographs obtained at different magnifications under environmental and high vacuum conditions, respectively. The EDS analysis confirms the purity of the compound (Table 3 and Table 4). The low vacuum mode measurement was performed with an uncoated sample while in the high vacuum mode the sample was coated with Au/Pd using a sputter coater. The surface morphology was analyzed through secondary electron imaging (SE), while the composition details were revealed by backscatter secondary electron imaging (BSED).

### 2.3. Computational Analysis

SwissADME web tool [33] and *ab initio* computations were used to estimate the lipophilicity of 12-crown-4, [Na(12-crown-4)_2_]^+^, [Na(12-crown-4)_2_]I_3_ and [Na(12-crown-4)_2_][(I_3_)_2_]^−^. The results are collected in Table 5. Additional information is provided in the Appendix A.

On the basis of the log *P*_o/w_ values (Table 5), [Na(12-crown-4)_2_]I_3_, I_2_, I^−^ and I_3_^−^ are lipophilic while the [Na(12-crown-4)_2_]^+^_,_ [Na(12-crown-4)_2_][I_3_]_2_^−^ and chitosan tend to be hydrophilic. When the I_3_^−^ anion is introduced to the [Na(12-crown-4)_2_]^+^ moiety, the lipophilicity increases. This can account for the complex to be membrane-permeable and release iodine for biological activity as a controlled release formulation mentioned by Harpal et al. [48]. After the release, the remaining compound [Na(12-crown-4)_2_]^+^ is hydrophilic and interacts with the hydrophilic, negatively charged bacterial cell wall for further action.

The lipophilicity of I^−^ and I_3_^−^ should lead to the same mode of action like iodine against all pathogens (Table 5) but molecular iodine remains still more potent than the anions [48]. Iodide and triiodide cannot iodinate fatty acids of the cell wall through symmetrical addition. Further, these ions cannot cross the cell membrane like the uncharged iodine molecule. The two ions pass through porin channels, which are available in Gram-negative bacteria. According to the calculated log *P*_o/w_ values, the triiodide should exert the highest antimicrobial effect, followed by iodine and finally, the iodide ion. The results in Table 1 reveal that iodine has stronger antimicrobial activity on all studied microorganisms than the iodide ion in NaI or the triiodide ion in [Na(12-crown-4)_2_]I_3_. At the same time, NaI has stronger action on Gram-negative bacteria compared to Gram-positive bacteria and *C. albicans* (Table 1). The iodide ion, with its lower lipophilicity and negative charge (Table 5), diffuses through the porin channels of the Gram-negative pathogens. 

The lipophilic compound [Na(12-crown-4)_2_]I_3_ inhibited the Gram-positive bacteria and the yeast *C. albicans* more than the iodide ion in NaI (Table 1). The higher lipophilicity of I_3_^−^ and [Na(12-crown-4)_2_]I_3_ should allow diffusion through cell membranes of Gram-positive bacteria. This is not possible, because the triiodide anion is charged and the compound [Na(12-crown-4)_2_]I_3_ is too large to diffuse through cell membranes. The large species [Na(12-crown-4)_2_]I_3_ and chitosan are not able to pass through these channels and engage in electrostatic/dipol-dipol interactions on the bacterial cell wall effectively [49]. The small, uncharged and lipophilic iodine molecule acts rapidly as broad-spectrum microbicidal agent against Gram-positive and Gram-negative bacteria [27,48,50]. Larger hydrophilic molecules with molecular weights less than 600–700 pass across the outer membrane through porin channels [51]. The large, lipophilic complex [Na(12-crown-4)_2_]I_3_ may, therefore, have dipole-dipole/electrostatic interactions with the bacterial cell wall of microorganisms and release the iodine, which penetrates the plasma membrane and directly attacks the proteins [31]. Subsequently, the remaining hydrophilic cation should exert its actions on the bacterial cell membranes with the ability to move through porin channels in Gram-negative bacteria and interacting with the peptidoglycan layer in Gram-positive pathogens.

12-Crown-4 is slightly hydrophilic according to the log *P*_o/w_ values and will not be able to diffuse easily through the cell membrane like I_2_. Small hydrophilic, ionic compounds pass through porin channels of Gram-negative bacteria. The two hydrophilic species 12-crown-4 and [Na(12-crown-4)_2_]^+^ are small enough to pass through the porin channels. This was confirmed through the results in Table 1. 12-Crown-4 inhibited Gram-negative bacteria, while Gram-positive bacteria and the yeast *C. albicans* were resistant because they have no porin channels. The larger species [Na(12-crown-4)_2_]I_3_ and chitosan are not able to pass through these channels. Instead, they interact with the bacterial cell wall [49].

In addition to these, we performed computations to investigate the energy required to release iodine from [I-I-I]^−^ and [I-I···I]^−^ as per processes **(1)** and **(2)** (see Table 6) using the geometries frozen to those obtained from experiment [52].

## 3. Discussion

Small, uncharged, lipophilic compounds, like iodine diffuse through the cell membrane of microorganisms. The microbicidal action of iodine is due to iodination of N-H, aromatic C-H and S-H groups in amino acids (methionin, cysteine), cytoplasm, lipids (unsaturated fatty acids) and microbial cell wall in the pathogens [6,27,31,48]. The iodination of aromatic C-H bonds (in tyrosine, histidine, uracil and cytosine) leads to steric hindrance by increasing the bulk of the molecule [6]. The iodination of S-H groups results in oxidation of the R-S-S-R disulfide groups leading to loss of connection between protein chains in the synthesis of proteins [6]. The iodination of unsaturated fatty acids changes the physical properties of the lipids [6]. The lipids double bonds are iodinated by symmetrical addition and the lipids change to saturated fats, which are known to solidify with their higher melting points and stiffer structure. This results in membrane immobilization [6]. Iodine has a strong microbicidal effect on pathogens by causing such irreversible damages [6,27,48,50]. They lead to denaturation of amino acids and nucleic acids in the microbial cell, inhibition of metabolic pathways and destabilization of the cell membrane structure [32,38]. This takes place through microbial membrane penetration, intracytoplasmic protein oxidation and iodination of cell proteins [6,31,32,48,50]. All the microorganisms studied are inhibited strongly by the antimicrobial activity of the effective antiseptic iodine with ZOIs of about 80 mm (Table 1).

Na(12-crown-4)_2_]I_3_ inhibited the same microorganisms with ZOI up to 40 mm (Table 1). This means, that the release of free molecular iodine is partly impaired by the strong halogen bonding within the three-center system [I-I-I]^−^ in the anionic structure.

Antimicrobial activities of free molecular iodine against *S. faecalis* and the Gram-negative *E. coli* were reported previously [1,27,48]. In this study, these microorganisms are also inhibited by iodine and the compound [Na(12-crown-4)_2_]I_3_ (Figure 2a, Table 1). The action of released free molecular iodine is obvious when antimicrobial activities of the triiodide complex [Na(12-crown-4)_2_]I_3_ and sodium iodide are compared. Our triiodide complex inhibits all the studied Gram-positive bacteria and *C. albicans* strongly, while the same pathogens are mostly resistant against NaI (Figure 2a, Table 1). This proves the release of free molecular iodine from the triiodide complex but in a reduced manner due to its strong halogen bonding. Gram-negative bacteria were more susceptible to NaI (Table 1). The antimicrobial effects of [Na(12-crown-4)_2_]I_3_ on the studied pathogens also depend on the chemical structure of the bacterial and fungal cell walls. 

Single-cell microscopic fungi like *C. albicans* have a cell wall and plasma membrane but their chemical composition is fundamentally different from those of bacteria [53,54,55]. The yeast *C. albicans* has negatively charged hydrophilic cell wall and plasma membrane. Antifungal agents have limited effectiveness because fungi have detoxification systems modifying the agents themselves and rendering resistance. Nevertheless, the few effective drugs target the membrane sterols or prevent their synthesis [53]. *C. albicans* has the highest susceptibility towards [Na(12-crown-4)_2_]I_3_ compared to all other selected bacteria. The triiodide compound is a strong antifungal agent against *C. albicans*.

The plasma membrane in Gram-positive bacteria is surrounded by the cell wall and consists of negatively charged peptidoglycans containing inclusions of teichoic acid and lipoteichoic acid [51]. The peptidoglycan forms a huge mesh-like structure with strands, which are cross-linked with each other. Most Gram-negative bacteria like *E. coli* connect the strands through the peptides directly and most Gram-positive bacteria, like *S. aureus* use a peptide interbridge instead [53]. Alternating amide groups form the chemical structure of the peptaglycine interbridge with partial negatively charged oxygen atoms [56]. Gram-positive bacteria have variable interpeptide bridges, length of the peptidoglycan strands and amount of cross-linking [56]. Gram-negative bacteria have considerably less cross-linking, with no inclusions of teichoic and lipoteichoic acid. Their outer membrane basically consists of lipopolysaccharides (LPS) that result in a higher negative charge on the cell surface than in Gram-positive microorganisms. This outer membrane of Gram-negative bacteria forms an additional barrier leading to more resistance against antimicrobial agents [51].

Large molecules like chitosan, chlorhexidine and [Na(12-crown-4)_2_]I_3_ cannot easily pass through the cell membranes and the protein channels used for transport and interact with the microbial membrane [48,49,51,53]. Such channels are present in the plasma membrane of Gram-positive bacteria and in both the outer and plasma membranes of Gram-negative bacteria [48]. The antibacterial activity of chitosan is due to an interaction between the positively charged chitosan (protonated amine groups) and the negatively charged bacterial membrane (LPS in Gram-negative bacteria and may be teichoic acid in the cell wall of Gram-positive bacteria) [50,51,53,57,58]. Similarly, the antimicrobial effect of [Na(12-crown-4)_2_]I_3_ is through the interaction between the partial positively charged carbon atoms in the crown ether complex and the negatively or partly negatively charged outer cell wall components of the microorganisms. 

### 3.1. Agar well and Disc Diffusion Assays

In our investigations of agar-well and disc diffusion assays, the Gram-positive pathogens were more sensitive to [Na(12-crown-4)_2_]I_3_ than the Gram-negative and this is possibly due to three reasons. 

Firstly, there are electrostatic interactions between the crown ether carbon atoms in the [Na(12-crown-4)_2_]I_3_ complex and the negatively charged teichoic and lipoteichoic acids on the Gram-positive bacterial cell wall, as well as the partial negatively charged oxygen atoms in the rest of the molecule including the peptide interbridges. These result in interactions, followed by deformation and partial destruction of the peptidoglycan structure due to the release of free molecular iodine into the thin periplasmic space and plasma membrane. Accordingly, *S. aureus* with its penta-glycine interbridges had the highest susceptibility towards [Na(12-crown-4)_2_]I_3_ compared with *E. coli* and the other selected Gram-negative pathogens, which all lack peptide interbridges. 

Secondly, according to the calculated log *P*_o/w_ values (Table 5), the triiodide complex is a lipophilic molecule and therefore, interacts effectively with or may even diffuse through the thick peptidoglycan layer of Gram-positive bacteria. This happens in the case of the lipid-soluble antibiotic chloramphenicol [51]. On the other hand, aminoglycosides, which are a class of hydrophilic antibiotics, are not very effective against Gram-positive microorganisms [51,53]. 

Gram-negative pathogens are sensitive to aminoglycosides because of their outer membrane structure. The outer membrane consists of an outer leaflet of negatively charged LPS and an inner leaflet of a layer of phospholipids with porin channels. The center of each porin protein monomer is a water-filled channel and allows passive diffusion of low molecular weight hydrophilic molecules. Therefore, the selected Gram-negative bacteria of this work were less sensitive towards the lipophilic compound [Na(12-crown-4)_2_]I_3_ (Table 1 and Table 5, Figure 1b). The lipophilic crown ether complex is not easily adsorbed on the hydrophilic Gram-negative bacterial cell wall (Table 5). Gram-positive bacteria were more sensitive to [Na(12-crown-4)_2_]I_3_ due to the ability of interaction (Figure 1, Figure 2 and Figure 3). 

Thirdly, molecular interactions lead to deformations within the structure. These deformations change the pure halogen bonding of the triiodide ion with its three-center-system [I-I-I]^−^ to [I-I···I]^−^ and the energy needed to release I_2_ is reduced (Table 6). The crown ether compound releases iodine and becomes a hydrophilic cation, ready to interact with the hydrophilic cell membranes. In Gram-positive microorganisms, iodine diffuses through the already damaged cell wall directly to enter into the cytoplasmic membrane within a short distance. In Gram-negative bacteria, iodine passes first through the bacterial outer membrane, then diffuses through the cell wall with thin layers of peptidoglycan, crosses through the periplasmic space and lastly, enters the plasma membrane. In Gram-negative and Gram-positive pathogens, iodine moves finally into the bacterial cytoplasm to oxidize the proteins [31,48]. 

In this context, the concentration of [Na(12-crown-4)_2_]I_3_ and its ease of interaction with the bacterial cell wall are directly linked to changes in the halogen bonding and its microbicidal properties due to the larger number of released free iodine molecules. Additionally, the availability or absence of interpeptide bridges, their length and polarity affect the dipol-dipol and electrostatic interactions. The crown ether compound interacts easily with the pentapeptide bridges, which are like loopholes within the mesh-like 3-dimensional structure increasing the flexibility of the peptidoglycan layer. Availability of interpeptide bridges is directly proportional to ease of interaction and release of free molecular iodine. The higher the concentration of released iodine with the shorter distance to reach the inner plasma membrane the higher is the antibacterial effect of the compound [Na(12-crown-4)_2_]I_3_. This was observed in the dilution series for the disc diffusion assays (Table 1 and Table 2). All effects together result in higher inhibition of Gram-positive *staphylococci* and *streptococci* by [Na(12-crown-4)_2_]I_3_ (Table 1). The hydrophobicity of the glucopyranoside rings of chitosan contributes to the insertion or translocation into the hydrophobic tail part of lipid [57]. The lipophilic [Na(12-crown-4)_2_]I_3_ follows similar mechanisms.

In our studies of well and disc diffusion assays, all the Gram-positive pathogens and *C. albicans* were either resistant or not sensitive towards 12-crown-4 when compared to the Gram-negative bacteria (Table 1). 12-crown-4 is hydrophilic and passes through porin channels of Gram-negative bacteria (Table 5). Gram-positive bacteria and the yeast *C. albicans* were resistant because they have no porin channels. The larger, lipophilic species [Na(12-crown-4)_2_]I_3_ cannot pass through these channels but instead interacts with their respective cell wall (Table 1 and Table 5). The highest sensitivity against 12-crown-4 occurred in the clinical isolate and reference strain of *K. pneumoniae*. The clinical isolate of *P. mirabilis* was resistant to the 12-crown-4 (Table 1). The assays for Gram-positive bacteria exhibited small inhibition zones, while the clinical isolates of *S. pneumoniae* and *E. faecalis*, as well as the fungi *C. albicans* were resistant to the crown ether. All the selected microorganisms were more susceptible to [Na(12-crown-4)_2_]I_3_ than to 12-crown-4. This proves that the triiodide compound has stronger interactions with the negatively charged cell membranes causing the release of free molecular iodine and resulting in microbial membrane damage. As a result, 12-crown-4 is not responsible for the antimicrobial action. The only possible agent remaining is the free molecular iodine. Iodine is released from [Na(12-crown-4)_2_]I_3_ after the triiodide with linear, symmetrical three-center-system [I-I-I]^−^ with strong halogen bonding changes due to interactions with microbial components to [I-I···I]^−^ units.

[Na(12-crown-4)_2_]I_3_ is a sandwich complex of two 12-crown-4 molecules surrounding one sodium ion. The size and molecular weight prevent the passage of this compound through the bacterial cell membrane [51]. Like chitosan, [Na(12-crown-4)_2_]I_3_ is attracted through electrostatic interactions. The partially positively charged carbon atoms in 12-crown-4 interact with the negatively charged LPS in the Gram-negative pathogens, as well as peptidoglycan, teichoic acid and lipoteichoic acid in Gram-positive *staphylococci* and *streptococci* [49,51,53,54,55,56,57]. Through these mechanisms, electrostatic interactions can lead to hydrolysis of peptidoglycans or even changes in the cell wall permeability due to osmotic differences resulting in inhibition of bacterial growth and leakage of intracellular components [49,51,53,54,55,56,57]. The disruption of microbial cell membranes changes bacterial metabolism and leads to bacterial cell death [49,51,53,54,55,56,57]. The Gram-positive microorganisms responded to [Na(12-crown-4)_2_]I_3_ more than to NaI, while *B. subtilis* and reference *E. faecalis ATCC 29212* showed opposite results. The release of free iodine from the stable triiodide units surrounding the complex cation is the main antimicrobial characteristic of the [Na(12-crown-4)_2_]I_3_ complex. The free molecular iodine penetrates the microbial membrane and leads to intracytoplasmic protein oxidation [48,49]. Sodium ions cannot enter through the thick peptidoglycan layer in Gram-positive microorganisms. They move by passive transport through porin channels in Gram-negative bacteria.

In general, the Gram-negative bacteria are slightly less affected by [Na(12-crown-4)_2_]I_3_ because of their stronger, outer barrier reducing such electrostatic interactions. *E. coli* and *P. aeruginosa* are inhibited more strongly by [Na(12-crown-4)_2_]I_3_ in comparison to the other Gram-negative bacteria, while *K. pneumoniae* is intermediately inhibited. NaI proves to be more effective through the attraction of the positively charged sodium ion to the negatively charged outer membranes of the Gram-negative microorganisms. This is the same mechanism of other ions entering the cell membrane of Gram-negative bacteria through cation-selective porins and causing toxicity [59]. The mechanisms of bacterial resistance include the occurrence of diminished protein channels on the bacterial outer membrane to decrease drug entry and/or the presence of efflux pumps to decrease the amount of drug accumulated within the cells [12,51,53]. 

The inhibitory action of [Na(12-crown-4)_2_]I_3_ on *E. coli*, *P. aeruginosa* and *K. pneumoniae* proves two possible mechanisms. After electrostatic interactions with the LPS, iodine and other components of the crown ether complex diffuse through the outer membrane. The diffusion leads to an increased influx of the toxic components into the cell through porin protein channels and/or a failure of the polyselective efflux pumps to eject the toxic compound from the intermembrane space [11]. An increasing number of strains produces efflux pumps in *Enterobacter aerogenes* and *K. pneumoniae* clinical isolates [11,51,53,60]. *P. aeruginosa* has a large number of efflux pumps capable of eliminating toxic compounds from the periplasm and cytoplasm [11,51,53]. This plays a key role in the multidrug-resistance of *P. aeruginosa* and happens usually in such a high rate that the drug concentrations are never enough to exert an antibacterial effect [11]. This action may have been impaired by the bacteriocidal effect of [Na(12-crown-4)_2_]I_3_ against *K. pneumoniae* and *P. aeruginosa*. 

Within the Gram-positive bacteria, the *staphylococci* were highly susceptible to [Na(12-crown-4)_2_]I_3_. *Staphylococci* are responsible for many diseases in humans and are found on mucous membranes and skin. *S. aureus* is a small round-shaped coccus which forms grape-like clusters, is non-motile and non-sporing [53]. [Na(12-crown-4)_2_]I_3_ inhibited both of the studied strains of *S. aureus* strongly. This is related to the ease of electrostatic interaction with the partial negatively charged groups in the teichoic acid, lipoteichoic acid and the interpeptide bridges. The clinical strain of *S. aureus* (ZOI = 39 mm) has a slightly smaller inhibition zone than the reference strain *S. aureus ATCC 25923* (ZOI = 43 mm). The next group which was highly inhibited are the *streptococci*. In this group, the most susceptible pathogen was *E. faecalis* (ZOI = 39 mm, Figure 3a). *E. faecalis* is a round-shaped coccus, appears singly, in pairs or short chains. It is non-motile and usually related to urinary tract infections (UTI) [51,53]. *S. pyogenes* was inhibited by the triiodide compound with a ZOI of 34 mm. *S. pyogenes* is also a round-shaped coccus, forms chains, is non-motile, non-sporing and aerotolerant [51,53]. The next bacteria with ZOI = 28 mm is *S. pneumoniae*. This microorganism is also a round-shaped coccus and appears in pairs. *S. pneumoniae*, also a *streptococcus*, is non-motile, facultative anaerobic and does not form spores as well [51,53]. 

The sequence of susceptibility follows an interesting pattern. The highest sensitivity towards [Na(12-crown-4)_2_]I_3_ occurs in *S. aureus*, followed by *E. faecalis* and *S. pyogenes*, followed by *S. pneumoniae*. All are round-shaped, non-motile cocci [53]. The susceptibility seems directly proportional to the complexity of the bacterial aggregation and inversely related to motility. Clusters are more sensitive to [Na(12-crown-4)_2_]I_3_ than chains and finally pairs. Cocci are more susceptible than rod-like bacilli.

The last Gram-positive bacteria with the smallest inhibition zone of 14 mm is *B. subtilis*, which is able to survive under harsh conditions and difficult environments [51]. It is a rod-shaped bacillus, which forms spores [53]. The Gram-negative bacteria *P. mirabilis* and *K. pneumoniae* show also intermediate ZOIs of 15 mm like *B. subtilis*. *K. pneumoniae* is a rod-shaped bacillus, which is facultatively anaerobic and non-motile but extremely resistant bacteria through encapsulation mechanism and mucoid activity [53]. It is found in the environment, as well as soil and surface water. *K. pneumoniae* colonizes human mucosal surfaces and spreads rapidly to other tissues leading to nosocomial infections in hospital settings, multidrug-resistance and high morbidity and mortality rates [53,61]. The Gram-negative bacteria *P. mirabilis* is a rod-shaped bacillus, which is facultatively anaerobic and has swarming motility [51,53]. This is contrary to the other motile Gram-negative bacteria, like *E. coli* and *P. aeruginosa*. *P. mirabilis* was most sensitive to iodine (ZOI = 64 mm), followed by gentamicin (35 mm), then NaI (22 mm), and finally, [Na(12-crown-4)_2_]I_3_ with 15 mm at a concentration of 13.3 mg/mL. The intermediate response to the crown ether complex is still remarkable. Among the Gram-negative bacteria, *E. coli* and *P. aeruginosa* have the highest susceptibility towards our triiodide compound (ZOI = 23 and 20 mm respectively). *E. coli* is a rod-shaped bacillus, which is motile and mostly found in the large intestine [53]. *P. aeruginosa* is also rod-shaped bacilli, which is motile, multidrug-resistant and a nosocomial pathogen, which is a major cause for morbidity and mortality [51].

Our compound [Na(12-crown-4)_2_]I_3_ inhibited the two motile Gram-negative bacteria *E. coli* and *P. aeruginosa* better than *P. mirabilis, K. pneumoniae* and the Gram-positive *B. subtilis*.

### 3.2. Antibiotic Screening Tests

The Gram-positive and Gram-negative bacteria were tested by disc diffusion method against commonly used antibiotics of different classes in comparison to 13.2 mM [Na(12-crown-4)_2_]I_3_ dissolved in methanol (Table 2). The bacteriostatic effect of the utilized antibiotic types erythromycin, gentamicin, streptomycin, chloramphenicol and amikacin is based on binding to different sites of bacterial ribosome preventing protein synthesis and replication [51,62,63]. They are all protein synthesis inhibitors. We performed our studies on clinical isolates and reference strains comparing [Na(12-crown-4)_2_]I_3_ with this class of antibiotics characteristic action. Our triiodide compound exerts at 13.3 mg/mL higher antibacterial activity against the Gram-positive microorganisms than the antibiotics used as positive controls (Figure 2b, Table 2).

Aminoglycosides, a class of hydrophilic antibiotics, includes gentamicin, streptomycin and amikacin [51,53,64]. They bind to the anionic compounds on the bacterial surface of mainly Gram-negative (LPS, phospholipids) and some Gram-positive (teichoic acid and phospholipids) pathogens [64]. These hydrophilic compounds diffuse through porin channels in Gram-negative microorganisms like the iodide- and triiodide-ion. This results in penetration into the periplasmic space and causes changes in the protein synthesis. Finally, this allows more molecules to enter and results in cell death [63,64]. The adsorption on the cell wall actually leads to cell wall depolarization. This change in the typical negative charge of the wall results in higher permeability and cell wall degradation [63]. Even electrostatic interactions with subsequent adsorption onto the bacterial cell wall may lead to its destruction by aggregation [65]. This aggregation causes cell envelope damage and changes the thickness and smoothness of the cell wall [65].

Gentamicin is bactericidal and a broad-spectrum antibiotic except against *streptococci* and anaerobic bacteria [51,53,63,64]. It is active against a wide range of bacterial infections caused by mostly Gram-negative bacteria including *P. aeruginosa*, *Proteus*, *E. coli*, *K. pneumoniae*. It is used in combination with other antibiotics (penicillin) to treat infections caused by Gram-positive bacteria like *S. aureus* and some species of streptococci. This drug diffuses across the cell wall to enter the bacterial cell and is often used with penicillin, which breaks down the cell wall to facilitate diffusion. The inhibitory action of gentamycin against *P. mirabilis* (35 mm), the clinical isolate *S. aureus* and the reference strain ATCC 25923 (both 30 mm) was stronger than the inhibition by [Na(12-crown-4)_2_]I_3_ (15, 18 and 0 mm, respectively). As a result, our triiodide compound is not able to diffuse through the cell wall of these related pathogens as effectively as Gentamycin.

Streptomycin inhibits the bacterial protein synthesis leading to the death of microbial cells. It is used as a broad-spectrum antibiotic against Gram-positive and Gram-negative bacteria [51,60,61]. The spore-forming bacteria *B. subtilis* was inhibited by streptomycin (ZOI = 22 mm), more effectively than by [Na(12-crown-4)_2_]I_3_ (10 mm). Our triiodide compound is not as effective as streptomycin in the inhibition of the protein synthesis in *B. subtilis*.

Amikacin blocks bacterial protein synthesis [51,60,61]. It is used as an antibiotic against multidrug-resistant, aerobic Gram-negative bacteria, especially *P. aeruginosa*, *E. coli*, *P. mirabilis* and *K. pneumoniae*. *E. coli* was inhibited by amikacin (ZOI = 22 mm), more efficiently than by [Na(12-crown-4)_2_]I_3_ (11 mm). Our triiodide complex is not blocking the protein synthesis in *E. coli* as effectively as amikacin. As a result, [Na(12-crown-4)_2_]I_3_ is not as effective against Gram-negative bacteria in diffusing through the cell membranes and blocking the protein synthesis as aminoglycosides (Table 2, Figure 2b).

Erythromycin belongs to the class of macrolides and is used to treat bacteria responsible for causing infections of the upper respiratory tract and skin by *Streptococcus* and *Staphylococcus* genera [51]. It has a bacteriostatic effect in higher concentrations, inhibiting the growth of bacteria. The inhibitory effect of erythromycin in comparison to [Na(12-crown-4)_2_]I_3_ was investigated by disc diffusion method in methanol on *S. pyogenes*, *E. faecalis*, *P. aeruginosa* and *K. pneumoniae*. The clinical strain of *S. pyogenes* was inhibited with a ZOI of 20 mm by [Na(12-crown-4)_2_]I_3_ more strongly than by erythromycin with 12 mm (Table 2). The other bacterial strains showed bigger ZOIs against erythromycin. The Gram-positive pathogen *E. faecalis* was inhibited by erythromycin (20 mm) slightly more than by the triiodide compound (ZOI = 15 mm), followed by the two Gram-negative bacteria *P. aeruginosa* (16 mm) and *K. pneumoniae* (13 mm) (Table 2). The inhibitory effect of [Na(12-crown-4)_2_]I_3_ compared to erythromycin is remarkable, especially in *K. pneumoniae* and *P. aeruginosa*.

Chloramphenicol has a broad-spectrum activity and is used for the treatment of infections caused by *S. aureus*, *S. pneumoniae* and *E. coli*, while it is not effective against *P. aeruginosa* [51]. Chloramphenicol is bactericidal and is not used unless in serious cases because it is toxic. It is lipid-soluble allowing it to diffuse through the bacterial cell membrane. Then, it binds reversibly to proteins of the ribosomal 50S subunit. In our studies, it was used against the reference strain of *S. pyogenes* (ATCC 19615) and exerted high ZOI of 30 mm compared to 16 mm by the triiodide complex. As a result, our compound [Na(12-crown-4)_2_]I_3_ was generally more effective against the studied Gram-positive pathogens compared to the macrolide erythromycin and the aminoglycoside gentamycin (Table 2, Figure 2b).

## 4. Materials and Methods 

### 4.1. Chemicals

Iodine (≥99.0%), potassium iodide, sodium iodide, dimethyl sulfoxide and Mueller Hinton Broth (MHB) were purchased from Sigma Aldrich (Gillingham, UK). Disposable sterilized Petri dishes with Mueller Hinton II agar and McFarland standard sets were purchased from Liofilchem Diagnostici (Roseto degli Abruzzi (TE), Italy). 1,4,7,10-Tetraoxacyclododecan (12-crown-4) was obtained from Sigma-Aldrich Chemical Co. (St. Louis, MO, USA). The bacterial strains *S. aureus* ATCC 25923, *E. faecalis* ATCC 29212, *S. pyogenes* ATCC 19615 were purchased from Liofilchem (Roseto degli Abruzzi (TE), Italy). *K. pneumoniae* WDCM00097 Vitroids, *E. coli* WDCM 00013 Vitroids, *P. aeruginosa* WDCM 00026 Vitroids and *C. albicans* WDCM 00054 Vitroids was bought from Sigma-Aldrich Chemical Co. (St. Louis, MO, USA). Erythromycin (SD013-1CT, 15 mcg/disc), gentamicin (SD170, 30 mcg/disc), chloramphenicol (SD081, 10 mcg/disc), streptomycin (SD031, 10 mcg/disc) and amikacin (SD035-1CT, 30 mcg/disc) were purchased from HiMedia Laboratories Pvt. Ltd. (Mumbai, India). Ethanol (analytical grade) was obtained from Fisher Scientific (Loughborough, UK) used as received. All other reagents were of analytical grade.

### 4.2. Optimized Preparation of [Na(12-crown-4)_2_]I_3_

0.09 g (0.63 mmol) NaI and 0.16 g (0.63 mmol) I_2_ are dissolved in 20 mL ethanol/10 mL methanol under stirring at room temperature. After heating to 40 °C, 0.2 mL (1.26 mmol) 12-crown-4 are added dropwise within 25 min under continuous stirring and heating to avoid precipitation. The clear solution is covered by parafilm and renders reddish-brown crystals of [Na(12-crown-4)_2_]I_3_ with a percentage yield of 70% after four days through slow evaporation at room temperature. The previous preparation method rendered only a yield of 59% because we used only 20 mL ethanol and added the 12-crown-4 at room temperature within 5 min [29].

### 4.3. Bacterial Strains and Culturing

All 10 microorganisms used in this study are from the clinical isolates collected from patients of Khalifa Hospital, Ajman, UAE. The strains were identified by standard biochemical methods. *S. aureus* ATCC 25923, *E. faecalis* ATCC 29212, *S. pyogenes* ATCC 19615, *K. pneumoniae* WDCM00097 Vitroids, *E. coli* WDCM 00013 Vitroids, *P. aeruginosa* WDCM 00026 Vitroids and *C. albicans* WDCM 00054 Vitroids were used as the reference strains. The strains were stored at -20 °C. All the selected strains were inoculated in MHB by adding the selected fresh bacteria and fungi into 10 mL Mueller Hinton broth. These suspensions were maintained at 4 °C until use. 

### 4.4. Investigation/Determination of Antibacterial and Antifungal Properties of [Na(12-crown-4)_2_]I_3_

[Na(12-crown-4)_2_]I_3_ was tested on Gram-positive *S. pneumoniae, S. aureus, S. pyogenes*, *E. faecalis*, spore-forming bacteria *B. subtilis* and Gram-negative bacteria *E. coli, P. mirabilis, P. aeruginosa* and *K. pneumoniae*. The antifungal activity of [Na(12-crown-4)_2_]I_3_ was tested on *C. albicans*. 

#### 4.4.1. Procedure for Zone of Inhibition Plate Studies

The antimicrobial activities of [Na(12-crown-4)_2_]I_3_ were evaluated against the above-mentioned microorganisms, using the zone of inhibition plate method because the crown ether complex was not completely soluble in polar media. Therefore, the method by Kirby-Bauer was selected [66]. The microorganisms were suspended in 10 mL Mueller Hinton Broth (MHB) and incubated for 2 to 4 h at 37 °C (bacteria) and 30 °C on Sabouraud Dextrose broth (fungi) according to microorganisms in use. Then a microbial culture (which has been adjusted to 0.5 McFarland standard), was used to lawn on the same agar plates evenly using a sterile swab. 100 μL was applied uniformly with sterile cotton swabs on disposable, sterilized Petri dishes containing Mueller Hinton II agar (MHA). These agar plates were dried for 10 min and then used for the agar well diffusion and disc diffusion methods.

#### 4.4.2. Agar Well Diffusion Method

A 6 mm diameter circular piece of MHB agar from an already inoculated agar plate was removed by a sterile well borer from the plate center and filled with 20 mg [Na(12-crown-4)_2_]I_3_. Similarly, three other agar plates were each loaded in the same way with 20 mg iodine, sodium iodide and 12-crown-4 as positive controls, while others were loaded with 75 µL of ethanol and 75 µL of methanol as negative controls. Then, these plates with the selected bacteria and the fungus *C. albicans* were incubated for 24 h at 37 °C and 30 °C, respectively. The diameter of the zone of inhibition was measured to the nearest millimeter with a ruler. The antimicrobial activity was evaluated based on the diameters of clear inhibition zone surrounding the well. The absence of an inhibition zone indicates no antimicrobial activity. The experiments were replicated three times and the results represent the average of three independent experiments.

#### 4.4.3. Disc Diffusion Method

Sterile filter paper discs (Himedia, India) with a diameter of 6 mm were impregnated with 2 mL of known concentrations of the stock solutions of [Na(12-crown-4)_2_]I_3_ in methanol/ethanol (10 mL of 17.63 mM concentration of the original concentration 13.3 mg/mL, 10 mL of 13.2 mM of 10 mg/mL and 10 mL of 5.29 mM of 4 mg/mL.)

Antimicrobial susceptibility to gentamicin, streptomycin, amikacin, erythromycin and chloramphenicol was performed by disk diffusion method according to Clinical and Laboratory Standards Institute (CLSI) recommendations [67]. Antibiotic discs were used against different microorganisms. Filter paper discs loaded with 2 mL of solvent (ethanol, methanol) were used as negative controls after they were dried in an oven at 37 °C for 30 min., placed on the previously inoculated agar plates and incubated at 37 °C for 24 h. The agar plates for *C. albicans* were incubated at 30 °C for 24 h. The diameter of the zone of inhibition was measured to the nearest millimeter with a ruler. The experiments were repeated three times and the results represent the average of three independent experiments. The antimicrobial activity was evaluated based on the diameters of clear inhibition zone surrounding the disc. The absence of an inhibition zone indicates no antimicrobial activity.

### 4.5. Scanning Electron Microscope (SEM) Analysis

The sample imaging and analysis were conducted on a tungsten-based Quanta ESEM (Environmental Scanning Electron Microscope, Fisher Scientific, Netherlands) at 5 and 10 kV. Through SE, BSED and EDS surface morphology, composition details and elemental EDS information were obtained, respectively by using two different modes. The low vacuum mode measurement was performed with an uncoated sample while in the high vacuum mode the sample was coated with Au/Pd using a sputter coater. 

### 4.6. Statistical Analysis

Data were expressed as mean ± standard deviation. Statistical significance between groups was calculated by one-way ANOVA. A value of *p* < 0.05 was considered to be statistically significant. All statistical analysis was performed using the SPSS software (version 17.0, SPSS Inc., Chicago, IL, USA).

## 5. Conclusions

Triiodides are promising candidates with a potential impact against growing antibiotic resistance. One of the major drawbacks of some iodine-containing disinfectants is a sublimation of iodine. This problem can be solved by introducing linear, symmetrical triiodides into the products like in the reported compound [Na(12-crown-4)_2_]I_3_. These “smart” triiodide ions with strong halogen bonding are a stable reservoir of I_2_-units which can be easily released after interaction with the pathogens. The linear, symmetric three-center-system [I-I-I]^−^ changes to an unsymmetrical triiodide [I-I···I]^−^. According to our computational analysis, the bond energy needed to release I_2_ from a [I-I···I]^−^ system is lower. By this way, free molecular iodine can be released directly on contact with the pathogen. Antimicrobial compounds with “smart” triiodides are more stable, release the iodine content in a more controlled way and have therefore longer durability. 

In this work, we related the biological activities and membrane permeability of the triiodide complex with estimated log *P*_o/w_. The results of the calculated values support our findings in the antimicrobial studies. 

The optimized synthesis of [Na(12-crown-4)_2_]I_3_ rendered a higher percent yield of product. The microstructural analysis by SEM confirmed purity, morphology, size, distribution, surface state and elemental mapping of this triiodide compound with its triclinic structure and previously published spectroscopic data. The antimicrobial activity of [Na(12-crown-4)_2_]I_3_ is based on the release of free molecular iodine and not related to 12-crown-4, nor iodide ions. 

The antibacterial activity of [Na(12-crown-4)_2_]I_3_ is directly proportional to the ease of interaction of the complex on the bacterial cell wall, its concentration and subsequent release of iodine, changing the previously lipophilic crown ether complex into a hydrophilic cation. Hence, [Na(12-crown-4)_2_]I_3_ is attracted through electrostatic interactions to the polar components of the microbial cell membranes. This interaction changes the halogen bonding within the symmetrical three-center-system from [I-I-I]^−^ to [I-I···I]^−^ and allows the release of free molecular iodine. Free molecular iodine penetrates immediately through bacterial membrane channels and causes oxidation of proteins within the bacterial cytoplasm as reported by several previous works. In general, the higher the bacterial aggregation in pairs, chains and clusters, the higher the sensitivity towards our crown ether complex. Non-motile, Gram-positive pathogens are more inhibited than Gram-negative, motile bacilli.

The present study investigated the bactericidal and fungicidal concentrations of [Na(12-crown-4)_2_]I_3_ against standard strains and clinical strains of microorganisms obtained from patients in the UAE. The triiodide compound inhibits all the 18 tested microbial strains. [Na(12-crown-4)_2_]I_3_ is a strong antifungal agent against *C. albicans*. It acts as a broad-spectrum bacteriocidal against the Gram-positive *S. aureus*, *S. pyogenes, E. faecalis*, and *S. pneumoniae*, as well as the Gram-negative pathogens *E. coli*, *P. aeruginosa* and *K. pneumoniae.* At a concentration of 13.3 mg/mL [Na(12-crown-4)_2_]I_3_ inhibits Gram-positive pathogens more than the selected antibiotics in disc diffusion studies. 

Future studies should investigate bonding patterns in polyiodides (halogen bonding/covalent bonding) and their influence on antimicrobial activity. Understanding this interdependence will be helpful in designing new, clinically relevant polyiodide compounds with enhanced inhibitory action against pathogens. 

## Figures and Tables

**Figure 1 pathogens-08-00182-f001:**
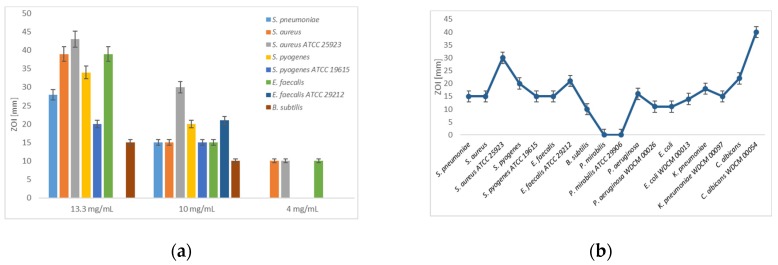
Disc diffusion assay of [Na(12-crown-4)_2_]I_3_: (**a**) Dilution series (13.3, 10 and 4 mg/mL) against Gram-positive bacteria.; (**b**) Dilution of 10 mg/mL against all studied microorganisms. The values are means from three repetitions and error bars are standard deviations (SD).

**Figure 2 pathogens-08-00182-f002:**
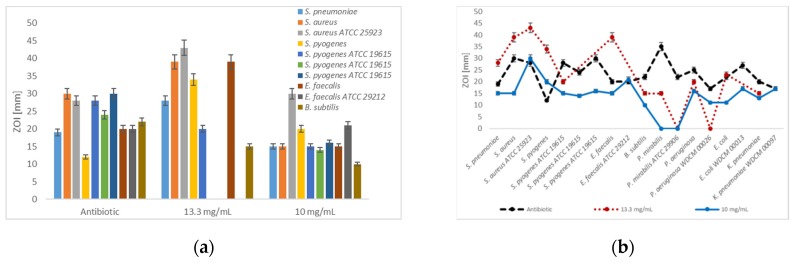
Disc diffusion assay-Dilution series of antibiotic positive controls and [Na(12-crown-4)_2_]I_3_ (**a**) Antibiotic positive control, 13.3 and 10 mg/mL against Gram-positive bacteria; (**b**) Antibiotic positive control, 13.3 and 10 mg/mL against all studied bacteria. The values are the mean from three repetitions and error bars are standard deviations (SD).

**Figure 3 pathogens-08-00182-f003:**
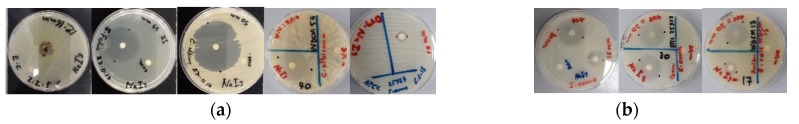
Antimicrobial agar plate methods on [Na(12-crown-4)_2_]I_3_. From left to right: (**a**) Agar well diffusion against *E. coli*, disc diffusion methods against *S. faecalis* and *C. albicans* (13.3 mg/mL), *C. albicans WDCM 00054* (10 mg/mL) and *S. aureus ATCC 25923* (4 mg/mL).; (**b**) Disc diffusion methods on [Na(12-crown-4)_2_]I_3_ (10 mg/mL) and positive control (antibiotic). From left to right: *S. aureus* clinical sample and gentamycin, *S. aureus ATCC 25932* and gentamycin, *E. coli WDCM 00013* and amikacin.

**Figure 4 pathogens-08-00182-f004:**
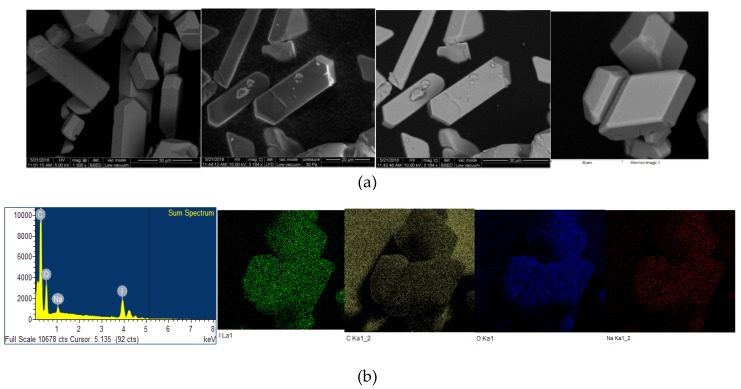
Microstructural analysis of uncoated [Na(12-crown-4)_2_]I_3_ at low vacuum mode. From up to down: (**a**) SE and BSED analysis.; (**b**) EDS analysis.

**Figure 5 pathogens-08-00182-f005:**
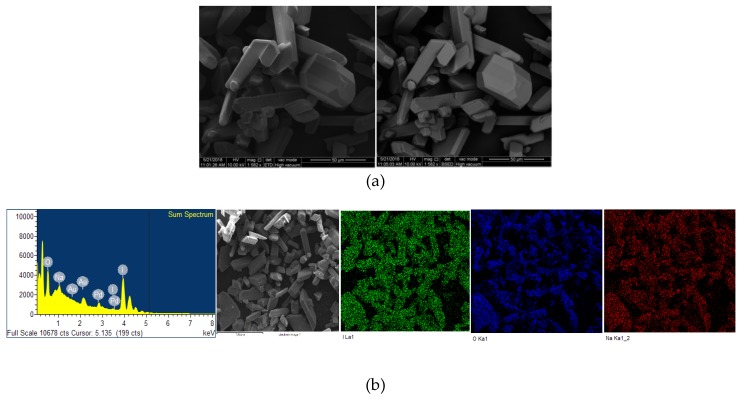
Microstructural analysis of Au/Pd coated [Na(12-crown-4)_2_]I_3_ at high vacuum mode. From up to down: (**a**) SE and BSED analysis.; (**b**) EDS analysis.

**Table 1 pathogens-08-00182-t001:** Antimicrobial activity of I_2_, NaI, 12-crown-4 (12C4) and [Na(12-crown-4)_2_]I_3_ (A) in Mueller Hinton Agar (MHA). ZOI [mm] against bacterial and fungal strains.

Strain	I_2_^[a]^	NaI^[a]^	12C4 ^[b]^	A^[a]^	A^[b] *^	A^[b] **^	A^[b] ***^
*S. pneumonia*	70	0	0	11	28	15	0
*S. aureus*	70	0	10	22	39	15	10
*S. aureus ATCC 25923*	70	0	0	-	43	30	10
*S. pyogenes*	60	0	9	15	34	20	0
*S. pyogenes ATCC 19615*	70	0	0	-	20	15	0
*E. faecalis*	80	0	0	16	39	15	10
*E. faecalis ATCC 29212*	52	25	0	-	-	21	-
*B. subtilis*	80	16	10	14	15	10	0
*P. mirabilis*	64	22	0	0	15	0	0
*P. mirabilis ATCC 29906*	47	20	0	-	-	0	-
*P. aeruginosa*	58	30	15	14	20	16	0
*P. aeruginosa WDCM 00026*	63	0	0	-	-	11	-
*E. coli*	58	27	12	16	23	11	0
*E. coli WDCM 00013*	60	19	14	-	-	14	-
*K. pneumoniae*	65	30	18	9	15	18	0
*K. pneumoniae WDCM 00097*	70	35	15	-	-	15	-
*C. albicans*	30	0	0	39	50	22	10
*C. albicans WDCM 00054*	80	0	0	-	-	40	-

[a] well diffusion studies (20 mg solid substance or 70 μL liquid substance in 6 mm diameter well). [b] disc diffusion studies (6 mm diameter disc impregnated with 2 mL of liquid 12-crown-4 or [Na(12-crown-4)_2_]I_3_ dissolved as * 17.63 mM (13.3 mg/mL), ** 13.2 mM (10 mg/mL), and *** 5.29 mM (4 mg/mL)). Grey shaded area represents Gram-negative bacteria. 0 = Resistant, - = not performed. No statistically significant differences (*p* > 0.05) between row based values through Pearson correlation.

**Table 2 pathogens-08-00182-t002:** Antibacterial activity of selected antibiotics (B) and [Na(12-crown-4)_2_]I_3_ (A). ZOI [mm] against bacterial and fungal strains.

Strain	Antibiotic	[mcg/disc]	B	A^[b] *^	A^[b] **^
*S. pneumoniae*	E	15	19	28	15
*S. aureus*	G	30	30	39	15
*S. aureus ATCC 25923*	G	30	28	43	30
*S. pyogenes*	E	15	12	34	20
*S. pyogenes ATCC 19615*	E	15 15	28 29*^+^*	20	15 14*^+^*
	G	30 30	24 27*^+^*	-	14
	C	10 10	30 27	-	16 14
*E. faecalis*	E	15	20	39	15
*E. faecalis ATCC 29212*	E	15 15	20 18^+^	-	21
*B. subtilis*	S	10	22	15	10
*P. mirabilis*	G	30	35	15	0
*P. mirabilis ATCC 29906*	G	30	22	-	0
*P. aeruginosa*	E	15	25	20	16
*P. aeruginosa WDCM 00026*	E	15	17	-	11
*E. coli*	A	30	22	23	11
*E. coli WDCM 00013*	A	30	27	-	17
*K. pneumoniae*	E	15	20	15	13
*K. pneumoniae WDCM 00097*	E	15	17	-	17

[+] blood agar. [b] disc diffusion studies (6 mm disc impregnated with 2 mL of * 17.63 mM (13.3 mg/mL), ** 13.2 mM (10 mg/mL) [Na(12-crown-4)_2_]I_3_)). E Erythromycin. G Gentamicin. C Chloramphenicol. S Streptomycin. A Amikacin. Grey shaded area represents Gram-negative bacteria. 0 = Resistant, - = not performed. No statistically significant differences (*p* > 0.05) between row based values through Pearson correlation.

**Table 3 pathogens-08-00182-t003:** EDS analysis of uncoated [Na(12-crown-4)_2_]I_3_ at low vacuum mode.

Element	C (K)	O (K)	Na (K)	I (L)	Total
*Weight%*	50.61	20.51	1.97	26.91	100
*Atomic%*	72.77	22.14	1.44	3.66	

**Table 4 pathogens-08-00182-t004:** EDS analysis of Au/Pd coated [Na(12-crown-4)_2_]I_3_ at high vacuum mode.

Element	C (K)	O (K)	Na (K)	I (L)	Au (M)	Pd (L)	Total
*Weight %*	50.61	20.51	1.97	26.91	10.13	6.41	100
*Atomic%*	72.77	22.14	1.44	3.66	3.10	3.63	

**Table 5 pathogens-08-00182-t005:** log *P*_o/w_ values of species relevant to this study.

Species	Consensus log *P*_o/w_ ^[a]^	log *P*_o/w_ (XLOGP3)	log *P*_o/w_	log *P*_o/w_ (*ab initio*)
I_2_	−	1.7 ^[b]^	2.49 ^[c]^	0.14 ^[e]^
I^−^	−	0.9 ^[b]^	−	4.04 ^[e]^
I_3_^−^	−	2.6 ^[b]^	−	2.69 ^[e]^
12-crown-4	0.59	−0.04 ^[a]^	−0.34 ^[d]^	1.66 ^[e]^
[Na(12-crown-4)_2_]^+^	−2.80	−1.55 ^[a]^	−	−
[Na(12-crown-4)_2_]I_3_	−1.87	1.03 ^[a]^	−	3.13 ^[f]^, 3.60 ^[g]^
[Na(12-crown-4)_2_][(I_3_)_2_]^−^	−1.30	3.61 ^[a]^	−	5.32 ^[e]^, 6.74 ^[f]^
Chitosan	−15.80	−21.40 ^[a]^^,^ ^[b]^	−2.62 ^[d]^	−

[a] SwissADME [33]. [b] ref [34]. [c] ref [35]. [d] VCCLab [36]. [e] full optimization using the B3LYP-D3/6-311G(d,p) method [37,38,39,40,41] using Gaussian 09 [42] running on SEAGrid [43,44,45,46,47]. [f] single-point computations using the B3LYP-D3/6-311G(d,p) method by using frozen geometry of corresponding X-ray structure [29]. [g] data corresponds to two different I_3_^−^ environments.

**Table 6 pathogens-08-00182-t006:** Formation of iodine.

Process		Bond Dissociation Energy (kJ/mol) ^[a]^
[I–I–I]^−^ → [I–I] + [I]^−^	**(1)**	135.0
[I–I•••I]^−^ → [I–I] + [I]^−^	**(2)**	124.3

[a] Obtained from single-point computations using the B3LYP-D3/6-311G(d,p) method.

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
