# Peer review of "“Smart” Triiodide Compounds: Does Halogen Bonding Influence Antimicrobial Activities?"

_pathogens, 2019, doi:10.3390/pathogens8040182_

Round 1

Reviewer 1 Report

The manuscript addresses an important and still current problem of searching for effective antimicrobial compounds. The Authors focused on triiodide compounds. The research methods were correctly selected and the experiment was carried out using many species of microorganisms, which positively affects the value of work. The results were presented in detail and carefully.

However, there are a few drawbacks in the manuscript that you should sleep before publishing:

The Authors write that they carried out a statistical analysis of the results obtained. However, in the figures and tables there are no markings indicating the presence or absence of statistically significant differences between the values. This must be completed.

The Discussion section is definitely too long - it should be shortened.

Materials and methods 4.4.1 - Why were prepared suspensions of bacteria and fungi incubated? What about the different growth rates of the microorganisms tested? Why wasn't the standard 0.5 McF suspension used? How did incubation affect the initial number of bacteria and fungi in suspension? How did you determine this number and why only for bacteria? 4.4.3 - Were the used antibiotic discs ready (from manufacturer) or prepared by the Authors?

Reviewer 2 Report

The manuscript of Z. Edis et al. describes the study of the triiodide complex Na[(12-crown-4)2]I3 as an antimicrobial agent against different strains of Gram + and Gram - bacteria and fungus. The activity is correlated to the structure of the complex as compared to other molecules containing I, and a thorough analysis is performed to elucidate the possible mechanism of action. The manuscript is well written and pertinent for publication, with some minor corrections:

I would suggest to organize by groups of Gram +, gram - and fungus in tables 1 and 2, since the following analysis of results is done by differentiating the action against these different types of microbes. In table 1 and 2: 0 means that bacteria were resistant to the antimicrobial agents, but I could not find an explanation for the - symbol. Were these conditions not tested? In this case, why? Figure 4: scale bars on the images should be visible. Table 5: the footnotes are not clear enough.

Author Response

Please see the attachment for the reviewers comments.

Thank you very much
